# MASLegalBench: Benchmarking Multi-Agent Systems in Deductive Legal Reasoning

## Abstract

Multi-agent systems (MAS), leveraging the remarkable capabilities of Large Language Models (LLMs), show great potential in addressing complex tasks. In this context, integrating MAS with legal tasks is a crucial step. While previous studies have developed legal benchmarks for LLM agents, none are specifically designed to consider the unique advantages of MAS, such as task decomposition, agent specialization, and flexible training. In fact, the lack of evaluation methods limits the potential of MAS in the legal domain. To address this gap, we propose MASLegalBench, a legal benchmark tailored for MAS and designed with a deductive reasoning approach. Our benchmark uses GDPR as the application scenario, encompassing extensive background knowledge and covering complex reasoning processes that effectively reflect the intricacies of real-world legal situations. Furthermore, we manually design various role-based MAS and conduct extensive experiments using different state-of-the-art LLMs. Our results highlight the strengths, limitations, and potential areas for improvement of existing models and MAS architectures.

## 1 Introduction

While LLM agents have demonstrated strong capabilities across numerous tasks (Anthropic, 2024; Yan et al., 2024), their performance can be limited when addressing complex problems. To overcome these limitations, multi-agent systems (MAS) composed of multiple LLM agents have attracted increasing attention from researchers (Ke et al., 2025a). MAS extends beyond simple agent-environment interactions by facilitating communication among agents. MAS typically consists of a Meta-LLM and multiple sub-agents. The Meta-LLM performs macro-level coordination, such as decomposing tasks for sub-agents and providing feedback on their outputs (Ke et al., 2025b). Each agent assumes a distinct role (Shinn et al., 2023) and exchanges messages with others. MAS have already achieved notable successes across multiple domains, including medicine (Li et al., 2025; Gawade et al., 2025), scientific research (Zhang et al., 2025), and social simulations (Yang et al., 2025; Kong et al., 2025).

The continuous development of MAS methods, coupled with their success in other domains, opens up new possibilities for legal tasks. In essence, MAS have several advantages that can be leveraged for legal reasoning. For example, their capability for task decomposition allows them to handle complex legal processes more effectively, which is often required in real-world scenarios. Additionally, MAS with role-based agents enable a structured division of labor that mirrors human collaboration in legal case handling. Unfortunately, few studies have explored the potential of MAS in legal tasks, and the absence of suitable evaluation methods constrains the successful transfer of MAS capabilities to the legal domain.

To bridge this gap, we aim to develop a MAS-adapted legal benchmark that leverages the key strengths of MAS. Our benchmark reflects deductive logic and the legal intuition involved in applying statutory provisions to specific factual scenarios. Our benchmark collects real court cases and proceduralizes their legal questions. Following a deductive reasoning paradigm with human verification, we extract a knowledge base primarily composed of facts, rules, their alignments, and common-sense inferences, along with a set of legal questions and their corresponding answers. Instead of merely relying on the case background, our benchmark provides a structured foundation that enables clearer and more principled agent specialization. We consider each reasoning step as

one of sub-tasks, including identifying the relevant legal rules and facts, establishing explicit correspondences between law and facts, leveraging common sense to infer additional relations, and ultimately deriving a well-grounded legal conclusion for the given issues. These subproblems can then be passed to the MAS, where the Meta-LLM collaborates with specialized agents to resolve them. To assess the potential of MAS in the legal domain, we manually configured a series of MAS systems and conducted extensive experiments.

Our contributions can be summarized as follows.

1) **Legal benchmark for MAS**. To the best of our knowledge, this is the first benchmark that provides sufficiently rich context to enable multiple LLM agents to collaborate in reasoning and exploration. Additionally, it is the first benchmark that allows MAS to distill problem decomposition directly from real-world legal cases. Our benchmark is built on expert-authored court cases, each supplemented with rich contextual details and comprising a total of 950 legal questions.

2) **Legal MAS designs**. We manually design a series of MAS tailored to our benchmark for executing legal tasks. These foundational MAS configurations enable us to validate the advantages of MAS over standalone LLM reasoning.

3) **Extensive experiments**. We conduct extensive experiments by varying MAS configurations and substituting different Meta-LLMs. The results demonstrate that introducing additional specialized agents enriches the available context, thereby enhancing LLM performance. Moreover, the experiments reveal notable inter-agent synergies: while individual agents may struggle when operating alone, their coordinated presence leads to substantially greater improvements.

## 2 PRELIMINARY

### 2.1 LEGAL REASONING

The study of legal reasoning has evolved through several principal paradigms. One line of work focuses on summarizing and structuring legal texts, making the content easier to understand for laypersons. Classic approaches include Legal Document Summarization (LDS) (Zhong & Litman, 2022; Shen et al., 2022) and Legal Argument Mining (LAM) (Santin et al., 2023; Palau & Moens, 2009). Another line of research emphasizes predictive modeling of new data, seeking to leverage historical information to generate insights for future scenarios. This line of research includes Legal Question Answering (LQA) (Zhang et al., 2023; Sovrano et al., 2020) and Legal judgments Prediction (LJP) (Huang et al., 2024; de Arriba-Pérez et al., 2022). Before the strong potential of LLMs was recognized, these tasks were typically framed as multi-class classification problems solved with classifiers.

Recently, with the rise of LLMs, the range of tasks and methods has expanded significantly, and their effectiveness has also been greatly improved. The powerful natural language capabilities of LLMs have inspired a range of tasks beyond classification, such as automated legal consultation (Cui et al., 2023) and contract generation (Wang et al., 2025). Subsequently, LLM agents have once again vitalized more complex forms of legal reasoning (Riedl & Desai, 2025), for instance ChatLaw (Cui et al., 2023), a multi-agent collaborative legal assistant.

### 2.2 EVALUATING LLMS IN LEGAL DOMAIN

As the potential applications of LLMs in the legal domain become increasingly evident, existing general-domain benchmarks fail to capture the full complexity and subtle nuances of real-world judicial cognition and decision-making. To address that, LawBench conducts evaluations from three perspectives: how LLMs memorize, understand, and apply legal knowledge (Fei et al., 2023). Legal-Bench is a collaboratively built benchmark that encompasses a wider variety of tasks and legal domains. What's more, the emergence of LLM agents has broadened the influence of LLMs within the law of agency. For example, Riedl & Desai (2025) discusses several under-theorized key issues, including questions of loyalty and the role of third parties interacting with agents. LegalAgentBench also offers a testing dataset specifically designed for LLM agent workflows (Li et al., 2024).

## 2.3 ENHANCE LEGAL REASONING WITH MULTI-AGENT COLLABORATION

LLMs generally encounter the following challenges in legal reasoning (Yuan et al., 2024): 1. Inconsistent reasoning. Legal reasoning typically requires multi-step, compositional logic (Servantez et al., 2024). However, LLMs are prone to distraction during intermediate reasoning steps (Shi et al., 2023). 2. Lack of grounding information. Legal provisions are often expressed in highly abstract terms, while real-world cases involve concrete and nuanced facts. Bridging this gap and aligning factual descriptions with legal concepts remains a major challenge. 3. Lack of domain knowledge. LLMs may hallucinate inaccurate legal knowledge or struggle with gaps in common-sense understanding (Dahl et al., 2024; Huang & Chang, 2022). Fundamentally, these challenges can be mitigated through task decomposition and role specialization, which are core principles of MAS.

Researchers have explored systems that incorporate auto-planners and sub-task agents to address these challenges (Yuan et al., 2024). However, the training of such systems often relies heavily on the correctness of the final outcome. To extend this line of research and provide a solid evaluation foundation for future legal MAS, we propose MASLegalBench designed specifically to support MAS.

## 2.4 IRAC METHOD

The IRAC method is a framework for organizing and structuring legal analysis, breaking down a legal question into four distinct steps: Issue (the legal question), Rule (the relevant law), Application (applying the law to the facts), and Conclusion (the final outcome)(IRAC Method, 2025). IRAC reasoning is designed to address the limitation of classical deductive reasoning, where the truth of the premises in a legal argument is often neither straightforward nor self-evident[1]. IRAC provides a logical framework for legal analysis as follows:

1) **Issue**. This is the legal question raised by factual ambiguity, resolved through precedent. For example, a filing deadline falling on a Sunday raises the issue of whether a Monday filing is timely.

2) **Rule**. It summarizes the legal principles relevant to the issue, distinguishing binding authority from persuasive sources.

3) **Application**. This applies the rules to the specific facts, explaining why each rule does or does not apply. This analysis, often considering both sides, is the core of IRAC, as it develops the answer to the issue.

4) **Conclusion**. It directly answers the issue without introducing new rules or analysis, restating the issue and providing the final determination based on the prior application of rules.

It should be noted that each IRAC step relies on the facts: issues are identified from the facts, rules are selected based on the facts, analysis interprets rules in light of the facts, and the conclusion applies the rules to the facts to resolve each issue.

## 3 TASK FORMULATIONS

### 3.1 EXTENDED IRAC REASONING

In this section, we refer to the IRAC method which is central to legal analysis. To address the lack of common-sense reasoning highlighted in Section 2.3, we extend IRAC by introducing Common Sense as a fifth component. Using this extended IRAC framework, any legal scenario can be systematically decomposed into these five components. With facts mentioned in Section 2.4, our task can be described as a deductive reasoning process revolving around six elements: to resolve an **Issue**, the MAS leverages **Facts** and relevant **Rules**, applies them through **Application**, and incorporates **Common Sense** to derive inferred relations that ultimately lead to the **Conclusion**. Figure 1 illustrates this process. Using IRAC analysis, when an MAS is tasked with addressing a legal question, it should follow the deductive reasoning steps outlined in Section 3.2.

---

[1]Nadia E. Nedzel, *Legal Reasoning, Research, and Writing for International Graduate Students* (New York: Aspen Publishers, 2021) `https://books.google.com.sg/books?id=4mVIzwEACAAJ`.

### 3.2 LEGAL MAS DESIGN

1) **Problem decomposition** Meta-LLM should first decompose the case $C$ into several potential domains, including the identification of the facts, the relevant rules, the application which is alignment of facts and rules, and the incorporation of common sense. This decomposition is performed recursively until each sub-task $s_t$ is atomic, meaning $s_t$ can be completed in a single reasoning step, making it more manageable for specialized agents to complete. This can be formed as Algorithm 1.

2) **Completion of sub-tasks**. Each sub-task should be handled by a specialized role-based agent, with different tasks being accomplished within distinct domains of knowledge. Following extended IRAC approach, we design four distinct role-based agents $[A_{facts}, A_{rule}, A_{analysis}, A_{commonsense}]$, each responsible for handling a specific reasoning step.

---

**Algorithm 1** Recursive Task Decomposition for Meta-LLM

---

1: **Initialize:** MetaLLM, Case introduction $C$, Guideline prompt for task decomposition $p_{template}$
2:     Sub-tasks queue $S_{queue}$, Sub-tasks results $S$
3: Compute sub-task set: $[s_{t_1}, s_{t_2}, \dots] = \text{MetaLLM}(C, p_{template})$
4: $S_{queue} = S_{queue} \cup [s_{t_1}, s_{t_2}, \dots]$
5: **for** each sub-task $s_{t_i}$ in $[s_{t_1}, s_{t_2}, \dots]$ **do**
6:     Evaluate $s_{t_i}$ for atomicity
7:     **if** $s_{t_i}$ is not atomic **then** $S_{queue} = S_{queue} \cup \text{MetaLLM}(s_{t_i}, p_{template})$
8:     **else** $S = S \cup \{s_{t_i}\}$
9:     **end if**
10: **end for**
11: **return** $S$

---

3) **Integration by the Meta-LLM**. After receiving the outputs from all sub-tasks, the Meta-LLM is responsible for integrating the results, supplementing any missing reasoning if necessary, and ultimately deriving the final conclusion.

Ultimately, the complete algorithm for a legal MAS can be summarized in Algorithm 2 .

---

**Algorithm 2** Legal MAS

---

1: **Initialize:** MetaLLM, Case introduction $C$, Guideline prompt for task decomposition $p_{\text{template}}$
2:     Guideline prompt for task accomplish $p_{\text{task}}$
3:     Role-based agents $A = [A_{facts}, A_{rule}, A_{analysis}, A_{commonsense}]$, Answer list $R = []$
4: Compute sub-task set: Sub-tasks results $S = $ Task Decomposition(MetaLLM, $C$, $p_{\text{template}}$)
5: **for** each sub-task $s_{t_i}$ in $S$ **do**
6:     Evaluating $s_{t_i}$ to the appropriate role-based agent $A_{t_i}$ from $A$
7:     Append $A_{t_i}(s_{t_i}, p_{\text{task}})$ to R
8: **end for**
9: **return** MetaLLM($C$,$R$)

---

## 4 MASLegalBenchmark

In this section, we present our choice of the General Data Protection Regulation (GDPR)[2] as the legal scenario. We collected real-world reports published by legal experts and extracted various types of knowledge from these reports to generate our benchmark.

### 4.1 DATA COLLECTION

Our data collection primarily focuses on simulating experts' problem decomposition processes and on capturing rich contextual knowledge to conduct deductive reasoning. To ensure our data contains the complete context, we gathered real GDPR court cases authored by experts, each of which provides a detailed and comprehensive account of a specific case. All data are sourced from the GDPR Enforcement Tracker[3], under the UK category. The original data are provided in PDF format with multi-level headings. We employed an LLM agent for PDF analysis, complemented by human checks, to construct a hierarchical tree structure for each document. For more details about the source data, please refer to Appendix A.

---

[2]https://gdpr-info.eu/
[3]https://www.enforcementtracker.com/

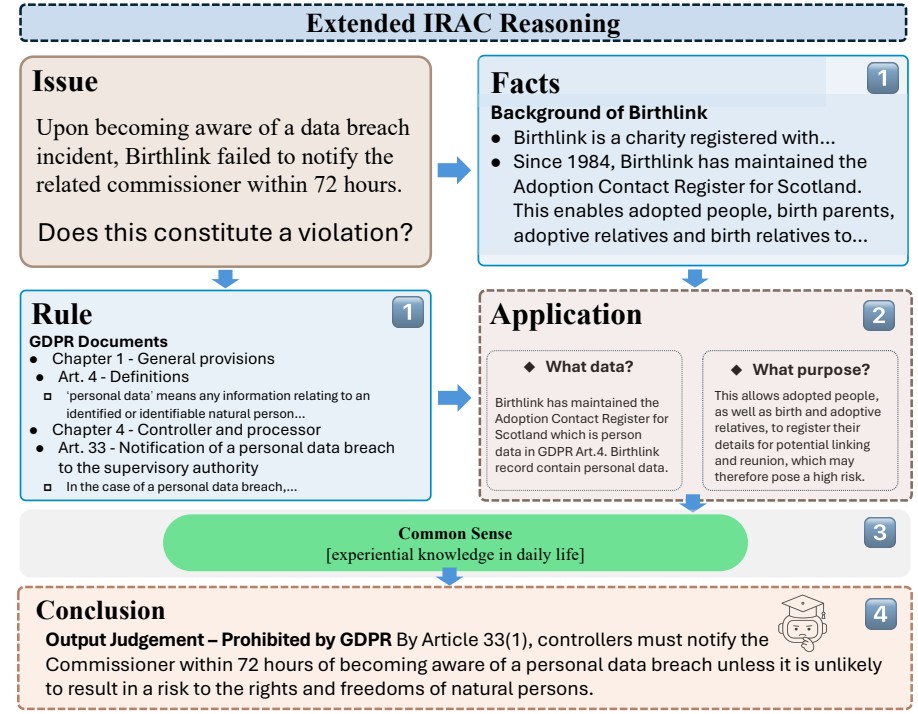

Figure 1: An overview of the enhanced IRAC reasoning process. Here, we take Birthlink (a company) as an example. In this case, a single issue is decomposed into several smaller questions, which are assigned to different agents: identifying the relevant facts and rules, inferring their alignment, and supplementing with common sense, before passing the results to the Meta-LLM for the final conclusion.

## 4.2 BENCHMARK CONSTRUCTION

After constructing the structure of each document, we relied on this hierarchical tree to identify the sections shared among all documents. We conclude that each document contains several sections, including at least an introduction, related legal framework, case background, legal nature of every entity, the commissioner's findings of infringement, final decision, calculation of penalties and an annex. Intuitively, we define the following mapping relations to bridge the actual data with our conceptual framework discussed in Section 3:

| Document Section | Mapping Type |
|---|---|
| Related legal framework | Legal rules |
| Case background | Reality facts |
| Legal nature of every entity | Application (Rule–Facts Alignment) |
| Commissioner's findings of infringement | Issue & Conclusion |

Furthermore, we consider the infringement findings of the commissioner as comprising two parts: the issues and the corresponding legal conclusion. From the collected reports, we aim to construct a set of legal questions, framing the problem so that, given reality issues as input, the Meta-LLM is tasked with generating the corresponding legal conclusion.

We employ DeepSeek-v3.1 to extract each issue from the reports and extract corresponding legal opinions as conclusion. In total, we construct 950 multiple choice questions (MCQs), comprising 647 yes/no questions and 303 single choice questions with four options each. For more details on our benchmark construction and statistics, please refer to Appendix B.

## 4.3 HUMAN EVALUATION

To verify the quality of the extracted sub-tasks, we conducted a human evaluation along the following three dimensions:

1) **Faithfulness**. Assesses whether the MCQs maintain semantic consistency to the original text.

2) **Clarity**. Assesses whether the extracted MCQs are expressed in a clear and unambiguous manner.

3) **Expertise**. Assesses whether the MCQs reflect appropriate legal expertise and professional depth.

The results are presented in Table 1. We invited three students with legal backgrounds or prior experience in legal-related research. A total of 30 samples were randomly selected, each including the original text, the extracted question, and the corresponding answer. Each sample was evaluated in three dimensions on a binary scale (0 or 1). This result (over 90% on every criterion) demonstrates that our benchmark consistently maintains high quality.

|  | **Faithfulness** | **Clarity** | **Expertise** |
|---|---|---|---|
| Evaluator 1 | 96.67 | 96.67 | 93.33 |
| Evaluator 2 | 100 | 100 | 100 |
| Evaluator 3 | 80.00 | 90.00 | 90.00 |
| **Average** | 92.22 | 95.56 | 94.44 |

Table 1: Human evaluation for our extracted benchmark. The results are reported in percentage form.

## 5 EXPERIMENT SETTINGS

In this section, we present the key experimental configurations. We manually designed a series of simple MAS setups, to systematically investigate the potential of MAS composed of role-based agents in the legal domain.

### 5.1 BENCHMARK SETUPS

As discussed in Section 4.2, our benchmark consists of two components: a predefined knowledge base containing facts, rules, and legal analyses that support the derivation of alignment and inferred relations, together with a set of MCQs that present issues and their corresponding conclusions. We aim to simulate the workflow described in Section 3.2, where a complex legal question is decomposed into a series of smaller elementary problems, each assigned to agents specialized in different reasoning steps. A RAG-based method is then employed to retrieve relevant outputs from these agents, assisting the Meta-LLM in generating the final answers. In practice, our experiments handle different steps in distinct ways. Specifically, since rules and facts are explicitly provided in the original data, we adopt a straightforward approach by directly leveraging the segmented source data to simulate the output of the corresponding agents. In contrast, application and common sense require additional processing of the original data, which is carried out by the corresponding agents. As a final judgments of the questions, Meta-LLM may generate answers (e.g., A, B, C, D, Yes, No) or produce a refusal response when the available context is insufficient. All prompt templates used for the agents and the Meta-LLM are provided in Appendix C.

We examine the performance of activating different sub-agents. In the following results, we use the abbreviations LR, F, AR, and CS to denote the activation of agents managing Legal Rules, Facts, Alignment Relations (Application), and Common Sense, respectively. The "+" symbol indicates the simultaneous activation of multiple agents. For example, LR+F+AR+CS represents the full deductive reasoning process, with agents from all four reasoning steps activated.

### 5.2 MODELS SELECTION

Our method adopts a RAG framework, where we implement two retrieval strategies: BM25 and embedding-based search (using the sentence-transformers/all-MiniLM-L6-v2 model (Hugging Face)). In our experiments, all agents designed for sub-tasks are implemented with DeepSeek-v3.1, while Meta-LLM explores a variety of leading open-source and closed-source models.

In the subsequent results, we report performance using the 'search method@hit' notation. For example, 'BM25@3' indicates that BM25 is used as the search method and retrieve the top-3 ranked outputs from sub-tasks, while 'EMB' indicates the use of embedding search.

### 5.3 BASELINES SELECTION

Since LR and F are directly provided in the original text, our agents do not perform additional processing beyond segmentation. Therefore, we select experimental groups containing only these

two steps as baselines, namely LR, F, and LR+F. Moreover, this set of baselines can also be regarded as purely RAG-based LLMs, highlighting the necessity of MAS collaboration. In addition, we report the precision of a fully random choice baseline without refusal, which is 42.03% listed in the first line of Table 2.

## 6 EXPERIMENT RESULTS

In this section, we conduct extensive experiments to evaluate the performance of the MAS series we designed on our benchmark.

| Meta-LLM | Activated Agents | Acc. (%) | | | | | |
|---|---|---|---|---|---|---|---|
| | | BM25@1 | BM25@3 | BM25@5 | EMB@1 | EMB@3 | EMB@5 |
| Random | None | 42.03 | – | – | – | – | – |
| Llama3.1-8B Instruct | F | 73.01 | **81.26** | 78.21 | 72.63 | 76.95 | 79.05 |
| | LR | **76.22** | 80.63 | 80.84 | 79.68 | 83.26 | 85.89 |
| | F+LR | 74.95 | 73.55 | 78.21 | 76.95 | 81.58 | 83.68 |
| | AR | 73.26 | 67.44 | 78.42 | 81.16 | **84.21** | 84.32 |
| | CS | **82.84** | 75.89 | **82.74** | **85.89** | **84.84** | **86.21** |
| | AR+CS | 72.84 | 77.58 | 78.11 | **82.52** | 82.84 | 83.26 |
| | F+LR+AR | 75.68 | **81.05** | **84.84** | 78.42 | 81.16 | **85.89** |
| | F+LR+AR+CS | 76.11 | 78.95 | 82.32 | 79.26 | 84.00 | 84.74 |
| Qwen2.5-7B Instruct | F | 53.79 | 60.42 | 63.47 | 57.16 | 64.11 | 68.53 |
| | LR | 58.95 | 62.95 | 70.63 | **68.00** | **73.47** | **76.95** |
| | F+LR | 60.53 | 64.95 | 68.63 | 62.00 | 69.58 | 72.42 |
| | AR | 52.53 | 52.74 | 58.84 | 66.95 | **72.74** | 74.42 |
| | CS | 62.84 | **69.79** | **73.16** | **69.37** | 72.00 | 74.53 |
| | AR+CS | 51.89 | 54.53 | 58.95 | 66.42 | 72.00 | 75.37 |
| | F+LR+AR | **62.94** | 66.00 | 70.84 | 64.84 | 70.11 | 74.21 |
| | F+LR+AR+CS | **62.74** | **67.26** | **72.95** | 64.95 | 70.95 | **75.47** |
| Qwen3-8B | F | 52.84 | 58.00 | 61.79 | 56.84 | 63.26 | 65.16 |
| | LR | 47.21 | 46.11 | 54.42 | 59.05 | 65.75 | **70.32** |
| | F+LR | 52.63 | 53.26 | 59.58 | 57.05 | **66.32** | **69.68** |
| | AR | 46.89 | 46.11 | 48.42 | **59.47** | 61.89 | 62.63 |
| | CS | 53.47 | 57.26 | 58.95 | 57.37 | 61.12 | 60.63 |
| | AR+CS | 46.05 | 46.84 | 50.16 | **59.79** | 61.01 | 62.59 |
| | F+LR+AR | **54.84** | **62.32** | 66.42 | 58.84 | **66.11** | 68.74 |
| | F+LR+AR+CS | **53.79** | **60.95** | **66.53** | 59.33 | 64.95 | 67.58 |
| DeepSeek-v3.1 | F | 34.00 | 45.47 | 50.53 | 44.84 | 56.21 | 59.26 |
| | LR | 28.84 | 37.37 | 46.32 | 50.11 | 57.05 | 59.79 |
| | F+LR | 36.84 | 38.32 | 46.32 | **52.32** | **60.95** | **64.32** |
| | AR | 24.00 | 30.21 | 34.84 | 41.79 | 45.26 | 48.42 |
| | CS | 29.58 | 34.95 | 39.37 | 39.37 | 40.00 | 42.95 |
| | AR+CS | 24.95 | 30.63 | 35.26 | 42.11 | 45.16 | 48.00 |
| | F+LR+AR | **40.84** | **52.11** | **56.84** | 52.21 | **60.21** | **63.05** |
| | F+LR+AR+CS | **40.95** | 51.89 | 54.42 | 52.00 | 59.79 | 62.53 |
| GPT-4o-mini | F | 65.05 | 73.89 | 78.00 | 70.00 | 77.37 | 79.68 |
| | LR | 57.37 | 70.21 | 79.79 | **76.95** | **82.84** | **84.32** |
| | F+LR | 66.74 | 72.32 | 78.32 | 73.26 | **81.58** | **82.95** |
| | AR | 61.26 | 67.26 | 70.32 | 74.00 | 79.16 | 79.16 |
| | CS | **70.32** | 76.11 | 76.21 | 74.95 | 78.21 | 79.89 |
| | AR+CS | 61.47 | 69.05 | 70.00 | 73.37 | 78.84 | 79.89 |
| | F+LR+AR | **70.84** | **78.63** | **82.06** | 74.11 | 80.84 | 82.42 |
| | F+LR+AR+CS | 70.00 | **78.63** | **81.58** | **75.16** | 80.95 | 81.89 |

Table 2: The results of different models on our benchmark vary across contexts and retrieval methods in legal judgment. **Bold-underlined** values indicate the context that yields the best performance, while **bold** values denote the second-best.

## 6.1 MAIN RESULTS

Our main results are presented in Table 2, which suggest the following findings:

1) *Richer contexts can lead to better performance.* The results indicate that involving more agents and providing richer reasoning steps generally leads to improved performance. For instance, 'BM25@5' outperforms 'BM25@3' when using the GPT-4o-mini model. Similarly, F+LR+AR+CS surpasses AR+IR with the DeepSeek-v3.1 model. This effect is more pronounced in larger-parameter models, such as DeepSeek-v3.1 and GPT-4o-mini, suggesting that the improvements brought by MAS enable the Meta-LLM to better evaluate the execution results of these agents.

Notably, while performance within the same context is nearly proportional to the number of retrieved chunks, the advantage of additional agents becomes less evident when comparing across different contexts. These results lead us to two preliminary insights: (1) enriched contexts with a greater number of agents generally enhance performance, and (2) the contributions of different agents vary, with their interactions remaining insufficiently understood.

2) *Our designed MAS demonstrates clear benefits in enhancing performance.* In Table 2, the shaded areas correspond to the MAS we designed, which extend the agents' capabilities to handle alignment relations and infer relations based on common sense. Our results show that 44 out of the 60 top performances (bold values) are achieved under our designed MAS, demonstrating the effectiveness of our MAS design as well as its potential for legal tasks.

3) *The best performance is often achieved when agents handling Legal Rules or Common Sense are activated.* From the best-performing results in the table, we observe that, with the exception of Llama3.1-8B-Instruct achieving its top performance under the F with BM25@3, all other peak results (bold-underlined values) occur in settings that include either LR or CS. This observation recalls the issue mentioned in Section 2.3, where LLMs may hallucinate regarding common sense and legal knowledge, highlighting the importance of carefully integrating MAS in legal reasoning tasks.

4) *When heavily relying on the outputs generated by agents, the Meta-LLM may often refuse to perform the task due to insufficient context.* Another notable finding is particularly evident with DeepSeek-v3.1 perform as Meta-LLM, whose accuracy under BM25 retrieval ranges only from 24.00% to 39.37%, even lower than random choice baseline. We conduct a case study on this phenomenon to investigate the underlying reasons, as illustrated in Figure 3. In this table, we report the proportion of cases where the Meta-LLM refused to provide an answer due to insufficient information. In the table, we observe that AR and AR+CS exhibit relatively high refusal rates, while F+LR+AR shows a lower refusal rate compared to F+LR. This indicates that activating AR agents may cause confusion and hinder effective judgment. This finding cautions that MAS should aim for collaborative integration of multiple agents rather than relying on a small subset of agents.

| Context | Refusal Rate (%) | | |
|---|---|---|---|
| | BM25@1 | BM25@3 | BM25@5 |
| F | 18.21 | 9.58 | 8.21 |
| LR | 19.37 | 12.84 | 10.11 |
| F + LR | 16.21 | 12.00 | 9.26 |
| AR | 22.32 | 16.63 | 14.00 |
| CS | 16.63 | 12.42 | 11.05 |
| AR+CS | 21.16 | 16.74 | 14.21 |
| F+LR+AR | 15.05 | 8.53 | 8.21 |
| F+LR+AR+CS | 15.68 | 8.42 | 8.32 |

Table 3: Refusal rates of DeepSeek-v3.1 across different configurations under BM25 retrieval.

## 6.2 AGREEMENT ACROSS DIFFERENT MAS CONFIGURATIONS

In this subsection, we focus on the interplay between agents by examining the agreement across different MAS configurations. We first use the results of MAS led by DeepSeek-v3.1 as an illustrative example. In Figure 3, we first compute the Cohen's Kappa agreement of DeepSeek-v3.1 under the BM25 retrieval setting across different configurations. Each cell reports the average agreement over 'BM25@1, @3, @5' under the same context and model. The three lowest pairwise agreements are highlighted in the figure with their values explicitly shown. The results show that agreement is lowest between 'LR systems' or 'F systems'. This finding motivated us to further investigate heatmaps

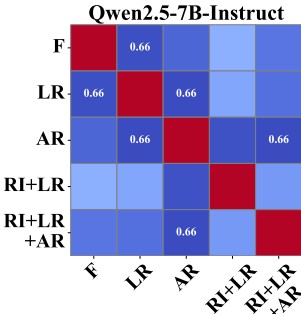 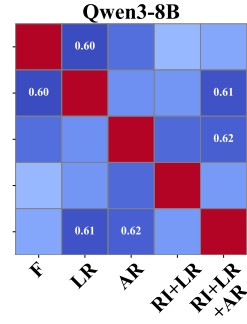 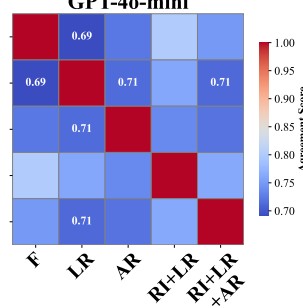

Figure 2: Heatmap of Cohen's Kappa agreement across individual knowledge types and models.

across multiple Meta-LLMs in Figure 2. These results reveal a common pattern: MAS with only LR and F tend to produce more inconsistent answers.

We then turn to an independent analysis of F, LR, F+LR, and F+LR+AR. Table 2 reveals a clearer trend under the BM25 retrieval method: performance generally follows the order F+LR+AR > F+LR > F/LR. When viewed through the lens of Figure 3 and Figure 2, this trend is further illustrated in the relatively high agreement between F+LR+AR and F+LR, as well as the consistently high agreement between F+LR and smaller systems (F and LR). In contrast, F+LR+AR shows noticeably larger discrepancies with F and LR. These results illustrate an iterative improvement process in MAS development, beginning with single-agent operations, progressing to dual-agent setups that collect both reality and legal knowledge, and culminating in more complex multi-agent systems that incorporate deductive reasoning. This process underscores the importance of collaborative interactions among multiple agents.

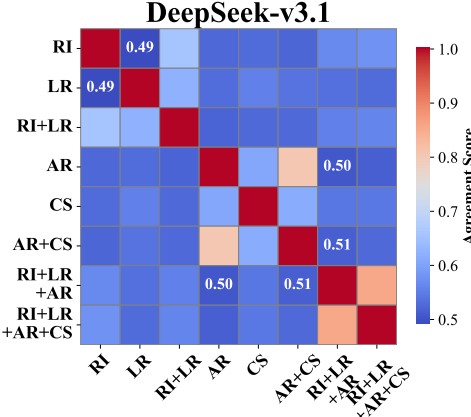

Figure 3: Heatmap of Cohen's Kappa agreement across different configurations for DeepSeek-v3.1 under the BM25 setting.

# 7    COMPARISON WITH EXISTING BENCHMARKS

To emphasize the distinctive features of our benchmark and our contributions, Table 4 compares current legal benchmarks for LLMs with our proposed benchmark. To better illustrate our motivation, the criteria here are specifically designed to accommodate MAS. There already exist many comprehensive benchmarks which cover a wide range of tasks and various areas of law. Our work serves as a complement to these benchmarks, and we present the following comparison.

| Benchmark Name | Taxonomy | Data Type | Task Decomposition | Real Data? | Fine-grained? |
|---|---|---|---|---|---|
| LawBench (Fei et al., 2023) | Fixed | Hibird | ✗ | ✓ | ✗ |
| LegalAgentBench (Li et al., 2024) | Fixed | Chinese law | ✗ | ✓ | ✓ |
| AgentsBench (Jiang & Yang, 2025) | Fixed | Criminal Law | ✓ | ✗ | ✗ |
| **MASLegalBench** (Ours) | Flexible | Court Cases | ✓ | ✓ | ✓ |

Table 4:  Comparisons among existing benchmark on LLMs.

# 8    CONCLUSION

In this study, to better leverage MAS for legal applications, we constructed the first benchmark tailored to the unique strengths of MAS, grounded in the deductive reasoning commonly used in legal analysis. To gain further insights, we developed a series of MAS designed to handle legal tasks and conducted experiments using these systems. The results indicate that the complex reasoning required in legal tasks and the adaptive interactions within MAS both point toward the tendency of multiple LLMs to collaborate through division of labor. However, a limitation of our work is that we do not consider automated MAS systems, which represent a major trend in MAS development.

ETHICS STATEMENT

We declare that all authors of this paper acknowledge the ICLR Code of Ethics. We generate the first benchmark on the integrity of MAS and legal tasks and a well-defined knowledge base based on publicly available enforcement reports from experts. During the download of relevant reports, we adhere to the official usage and access rules of the GDPR Enforcement Tracker[4]. Human evaluations and annotations are conducted by three students with legal backgrounds or prior experience in legal-related research to ensure the quality of the synthetic benchmark. Annotators are compensated at a rate of 15 USD per hour, above the local minimum wage. To the best of our knowledge, this work fully complies with open-source agreements.

Furthermore, we believe our benchmark can serve as a valuable asset for existing applications of LLMs in legal domain by advancing the application of MAS in the legal domain.

REPRODUCIBILITY STATEMENT

To ensure the reproducibility of our experimental results, we put our detailed implementations under Section 5. We also provide details about the source data and our benchmark in Appendix A and Appendix B. All the prompt templates used in our experiments are listed in Appendix C. Our reproducible code is also submitted as the Supplementary Materials. We will open-source the reproducible data and code.

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

# A    SOURCE DATA DESCRIPTIONS

## A.1    SOURCE DATA STATISTICS

Our source files are provided in PDF format, from which the dataset is constructed using publicly available GDPR enforcement cases. In total, it contains 15 distinct cases. Each case document ranges from 30 to 153 pages, with an average length of 59.80 pages. After preprocessing and segmentation, each case document was divided into a set of minimal text chunks, ranging from 67 to 439 per file, with a average of approximately 185.53 chunks. This granularity ensures manageable input sizes for downstream retrieval and reasoning tasks. Table 5 presents the length of each section from the source documents, quantified by the number of chunks, which serves as the basis for subsequent analysis.

| Section | Min | Max | Average |
|---|---|---|---|
| Introduction | 3 | 18 | 7.13 |
| Legal Framework | 2 | 55 | 18.27 |
| Background | 2 | 86 | 25.27 |
| Nature | 2 | 100 | 18.29 |
| Infringements | 14 | 129 | 59.27 |
| Decision | 1 | 118 | 28.93 |
| Penalty | 2 | 61 | 22.40 |
| Annex | 6 | 31 | 8.46 |

Table 5: Section length statistics in source documents (measured in chunks)

## A.2    SOURCE DATA SAMPLE

Similar to Figure 1, we provide an illustrative example using the original Birthlink case file[5]. The following illustrates the agenda structure of a source case document. The number on the right indicates the starting chunk index of each section.

---

[5]The source link for Birthlink reports: *https://ico.org.uk/media2/bvljtpy2/birthlink-mpn.pdf*

### A.3 MAPPING OF SOURCE DATA AND IRAC METHOD

This appendix illustrates how each section of the source case documents corresponds to elements of the IRAC reasoning framework. Chunk numbers indicate the starting position of each section, and sub-sections are mapped to specific reasoning steps. As mentioned in Section 4.2, each section is mapped to the corresponding IRAC elements, establishing a clear relationship between the source data and the deductive reasoning process.

Such a mapping allows us to systematically analyze how legal analysis is structured within each case, and how different reasoning steps are distributed throughout the document. By examining the chunk positions and section lengths, we can observe patterns in how legal arguments are developed, which sections tend to be more densely packed with rules versus facts. Analysis of the distributions shows that **Rule** sections are highly concentrated: they contain few chunks but carry key legal reasoning. In contrast, sections like **Background** and **Infringements** are larger, capturing detailed facts. This pattern indicates that in real-world cases, rules are concise yet critical, guiding the application and inference steps in the IRAC process.

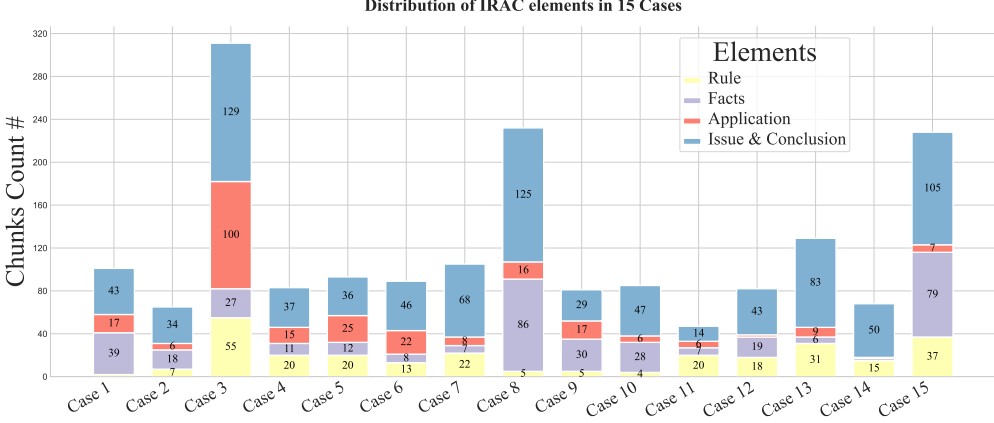

Figure 4: IRAC elements distribution across 15 cases. Each bar represents a case and is colored according to IRAC elements.

## B BENCHMARK DETAILS

### B.1 BENCHMARK CONSTRUCTION

We extract questions corresponding to the 'Issue & Conclusion' sections using DeepSeek-v3.1. During this process, we aim to preserve the original meaning of the text as much as possible, ensuring that the extracted questions faithfully reflect the legal reasoning presented in the case. The prompts used for this extraction are provided in Table 6. Additionally, to obtain more analytical data, we first determine whether the original text contains a legal decision. Based on this determination, we then categorize and extract questions accordingly, allowing us to differentiate between decision-based questions or opinion-based questions (non-decision based questions).

### B.2 BENCHMARK STATISTICS

Here, we provide additional benchmark data in Figure 5 and Figure 6, distinguishing questions along two dimensions: (i) whether they are answered in a binary form (yes/no) or multiple choice (a–d),

and (ii) whether they involve a legal decision (as opposed to a legal opinion). Whether a question involves a legal decision was determined during the benchmark construction process. We consider that questions containing a legal decision tend to have more definitive answers, whereas questions without a decision typically reflect legal opinions, which may introduce some ambiguity.

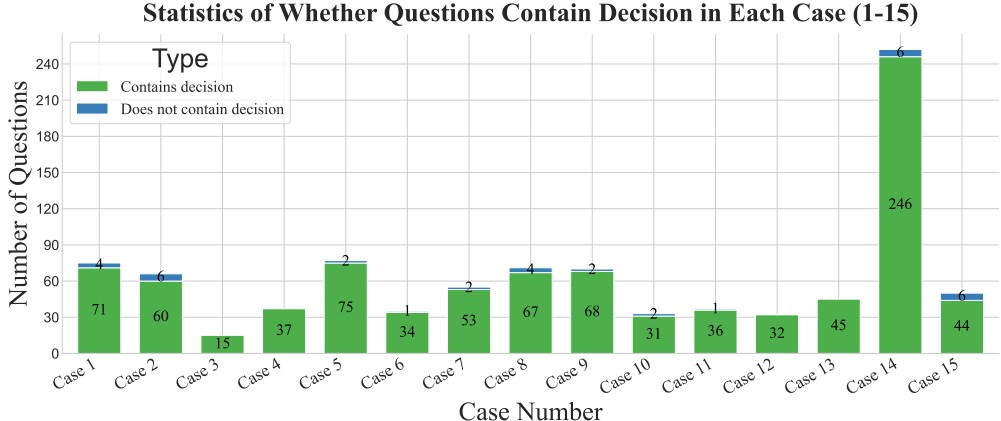

Figure 5: yes/no and abcd question distribution across 15 cases. Each bar represents a case and is divided by question type.

**Statistics of Whether Questions Contain Decision in Each Case (1-15)**

Figure 6: Question distribution across 15 cases based on whether they contain a decision. Each bar represents a case and is divided by question type.

## B.3 BENCHMARK SAMPLE

Here, we provide sample benchmark questions along with their corresponding response. As in Figure 1, we use the Birthlink case as an illustrative example.

---

**Benchmark Sample Question**

**Legal Decision:** Yes

**Question:** A company destroyed linked records through its processing activities, failing to ensure appropriate security of personal data including protection against unauthorized destruction. Does this constitute a violation of GDPR?

**Options:**

- Yes
- No

**Correct Answer:** Yes

---

---

**Benchmark Sample Question**

**Legal Decision:** Yes

**Question:** Did Birthlink violate Article 33(1) UK GDPR by failing to notify the Commissioner of a personal data breach within 72 hours of becoming aware of it and failing to implement appropriate measures to establish whether a breach had occurred?

**Options:**

- Yes
- No

**Correct Answer:** Yes

---

**Benchmark Sample Question**

**Legal Decision:** Yes

**Question:** When processing highly sensitive personal data that includes irreplaceable sentimental items, what level of security measures must an organization implement according to GDPR Article 32?

**Options:**

- A. Basic security measures appropriate for the risk level
- B. No specific security measures are required for charitable organizations
- C. Appropriate technical and organizational measures to ensure a level of security appropriate to the risk
- D. Security measures only if explicitly requested by data subjects

**Correct Answer:** C

---

**Benchmark Sample Question**

**Legal Decision:** Yes

**Question:** An organization processes highly sensitive personal data including sentimental items like handwritten letters and photographs for charitable purposes. The Commissioner finds that the nature of this processing, without appropriate security measures, was likely to result in high risk to data subjects. Which GDPR principle is most directly violated in this scenario?

**Options:**

- A. Principle of data minimization (Article 5(1)(c))
- B. Principle of integrity and confidentiality (Article 5(1)(f))
- C. Principle of purpose limitation (Article 5(1)(b))
- D. Principle of lawfulness of processing (Article 6)

**Correct Answer:** B

---

## C PROMPT TEMPLATES

### C.1 PROMPT FOR BENCHMARK CONSTRUCTION

In Section 4.2, we employ DeepSeek-v3.1 to assist in extracting legal questions from the source data, including both issues and corresponding conclusions. This approach allows us to systematically transform complex case documents into structured question–answer pairs suitable for benchmarking. The corresponding prompt templates used for guiding DeepSeek-v3.1 during this extraction process are provided in Table 6.

## C.2 PROMPT FOR AGENTS

### C.2.1 PROMPT FOR APPLICATION AGENTS

In Section 5.1, we employ specialized agents to extract Application relations directly from the source case documents. This process simulates how a real agent would gather and organize information from historical cases. The corresponding prompt templates used for guiding these agents are provided in Table 7.

### C.2.2 PROMPT FOR COMMON SENSE AGENTS

We utilize agents to extract inferred relations based on common sense from the existing source data. This approach simulates how an agent can leverage general reasoning and domain knowledge to derive additional alignments that are not explicitly stated in the text. The prompts designed for these Common Sense Agents are provided in Table 8

## C.3 PROMPT FOR META-LLM

By aggregating the outputs from multiple agents along with the original issue, the Meta-LLM is expected to identify the most convincing conclusion. This step simulates a human expert synthesizing diverse sources of information to reach a reasoned judgment. The prompt used to guide the Meta-LLM in this reasoning process is presented in Table 9.

You are an GDPR Commissioner.
Your task is to **rewrite the text into a MCQ question**.
Instructions:
1. Rewrite the text into a MCQ question, you should mainly focus on the following aspects:
    - Whether a behaviour or decision **violates the GDPR** or is **lawful**.
    - If the facts is not enough to be considered as a violation, you should consider it a behaviour not violated
       the specific regulation and also rewrite it into a question.
    - References to explicitly mentioned legal provisions or articles.
    - If the text **does not contain any facts or behaviours which can be used to judge whether the controller has
       infringed the GDPR**, return an empty list: [].
    - The question should not be reference to the original text.
2. If a fact or behaviour is present, rewrite it into a **self-contained MCQ question** **based on the factual
scenario**:
    - Present the **facts clearly**: who did what, how they did it, and under which circumstances.
    - Include **relevant legal provisions or articles**, if mentioned.
    - The question can sometimes switch between affirmative and negative forms of a statement, ex. ...is vio-
       lated... $->$ Yes can be switch to ...is comply... $->$ No.
    - Just use the name appeared in the text, ex. use 'company name' instead of 'Data controller'
3. There can be 2 or 4 options in the MCQ question.
    - If there are 2 options, the options are Yes and No.
    - If there are 4 options, the options are A, B, C, D and the correct answer is one of them.
    - Only one option should be correct.
4. Generate all the possible questions based on the factual scenario and provide them in the 'questions' field.
5. The JSON must be valid and properly formatted.
### Output JSON Format:
```
[
   {
      "whether_contains_decision": "true/false",
      "question": "....",
      "options": {
         "A": "...",
         "B": "...",
         "C": "...",
         "D": "..."
      },
      "correct_answer": "A/B/C/D"
   },
   {
      "whether_contains_decision": "true/false",
      "question": "....",
      "options": {
         "Yes",
         "No"
      },
      "correct_answer": "Yes/No"
   }
]
```
### Input:
{content}

Table 6: This prompt is designed for the benchmark construction to extract issues and their corresponding conclusions from legal case texts, and to convert them into a multiple-choice question (MCQ) format for evaluation purposes. Light blue text inside each "{}" block denotes a replaceable string variable.

You are an expert in GDPR.
Your task is to extract **alignment relationships** between real entities or concepts (companies, organisations, charities, regulators, data, records, etc.) and their legal roles or definitions under the GDPR.
Instructions:
1. Identify all **entities** (e.g., companies, charities, regulators) and **concepts** (e.g., filing system, personal data, special category data) **if explicitly extractable from the text**.
2. For each entity/concept, determine if the text assigns:
   - A **legal role** (Controller, Processor, Supervisory Authority, Data Subject), OR
   - A **legal classification/definition** (Filing System, Personal Data, Special Category Data).
   If no alignment can be extracted, do not include an entry.
3. Include the corresponding legal source only if explicitly mentioned; Otherwise, set legal_source to null.
4. Extract relations between entities/concepts only if explicitly stated (e.g., "X is stored in Y"); leave empty if none. Do not infer new relations.
5. Provide a short rationale for each item without referencing the original text.
6. The JSON must be valid and properly formatted.
### Output JSON Format:
{
    "entities_and_concepts": [
        { "entity_or_concept": "...", "legal_alignment": "...", "legal_source": "... or null", "rationale": "..." }
    ],
    "relations": [
        { "source": "...", "relation": "...", "target": "...", "rationale": "..." }
    ],
}
### Input:
{content}

Table 7: Prompt template for extracting application relations by agents. Light blue text inside each "{}" block denotes a replaceable string variable.

You are an expert in GDPR.
Your task is to extract inferred relationships between real entities or concepts (companies, organisations, charities, regulators, data, records, etc.) based on common sense.
Instructions:
1. Identify all entities (e.g., companies, charities, regulators) and concepts (e.g., filing system, personal data, special category data) if they can be explicitly extracted from the text.
2. For each entity or concept, determine if the text explicitly assigns:
   - A legal role (e.g., Controller, Processor, Supervisory Authority, Data Subject), OR
   - A legal classification/definition (e.g., Filing System, Personal Data, Special Category Data).
   If no alignment can be extracted, do not include an entry.
3. Extract relations between entities/concepts only if explicitly stated (e.g., "X is stored in Y"); leave empty if none.
4. Include an inferred_alignments section only if strictly derivable from existing alignments and relations; otherwise, leave empty.
5. Provide a short rationale for each item without referencing the original text.
6. The JSON must be valid and properly formatted.
### Output JSON Format:
{
    "inferred_alignments": [
        { "entity_or_concept": "...", "legal_alignment": "...", "legal_source": "...", "rationale": "..." }
    ]
}
### Input:
{content}

Table 8: Prompt template for extracting inferred alignments by agents. Light blue text inside each "{}" block denotes a replaceable string variable.

You are an expert in law.
Your task is to carefully read the given **context** and **question**, then provide the answer in JSON format.
Requirements:
1. The JSON must contain two fields:
   - "rationale": a short reasoning process explaining why this answer follows from the context.
   - "answer": the final concise answer to the question.
2. The reasoning should be **based only on the provided context**, without adding external knowledge unless strictly necessary.
3. The JSON must be valid and properly formatted.
### Output JSON Format:
{
   "rationale": "...",
   "answer": "..." (select from A/B/C/D/Yes/No)
}
Question: {question_content}
Context: {context}

Table 9: Prompt template for the Meta-LLM to generate conclusion answers based on questions and agents-provided context. Light blue text inside each "{}" block indicates a replaceable string variable.

