# OpenReview forum: "MASLegalBench: Benchmarking Multi-Agent Systems in Deductive Legal Reasoning"
_ICLR.cc/2026/Conference — ICLR 2026 Conference Withdrawn Submission_

### Official Review · Reviewer_W2TS · 2025-10-20

**Soundness:** 2
**Presentation:** 3
**Contribution:** 1
**Rating:** 2
**Confidence:** 4

**Summary:**

Multi-agent systems promise increased accuracy in solving complex tasks compared to vanilla LLMs. The authors claim there are no benchmarks yet that are suitable for measuring MAS performance in the legal domain. This is why they create MASLegalBench. MASLegalBench uses court cases tackling GDPR issues. The authors used an LLM to extract the sections from the PDFs and validated it with human checks. They ran a human evaluation to verify the quality of the extracted tasks. They find that 44 out of the top 60 performances were achieved by MAS compared to 16 by vanilla LLMs.

**Strengths:**

- The authors are early in evaluating MAS in the legal domain
- The authors pick a realistic evaluation domain that seems challenging
- The dataset construction and validation seems sound

**Weaknesses:**

- The focus on GDPR makes the benchmark rather narrow. Therefore the general title is a bit misleading and should probably be adjusted.
- To me it is not clear why the authors limited the retrieval to 5 hits in the largest case, since this appears like quite a low number given today's LLM context windows
- Both sparse and dense retrieval have their pros and cons, which is why they are usually combined. However, this experiments is missing
- Out of the five evaluated models only two are reasoning models (Qwen3-8B and DeepSeek-v3.1) and it is not clear what settings were used for inference. To me it is unclear why the authors don't focus their evaluation more on reasoning models given that they are benchmarking "Deductive Legal Reasoning".
- The name MASLegalBench is somewhat unfortunate since it suggests a connection to the LegalBench evaluation, but it operates on different data.

**Questions:**

- L065: what do you mean with expert-authored court cases?
- You might be interested in LEXam: Benchmarking Legal Reasoning on 340 Law Exams
- check the grammar on L302-303
- L473: typo in Data Type Hybrid
- What was the reasoning budget for the reasoning models?
- What are the inference settings you used (temperature, top_p, etc.)?

---

### Official Review · Reviewer_Pdab · 2025-10-30

**Soundness:** 1
**Presentation:** 3
**Contribution:** 2
**Rating:** 2
**Confidence:** 4

**Summary:**

This paper studies LLM agents for legal reasoning. First, a benchmark dataset is constructed, focused on GDPR law, containing several multiple choice questions regarding law cases. Afterwards, a MAS system is constructed, using LLM and non-LLM agents, which are aggregated by a meta-LLM agent. That system is then evaluated on the proposed benchmark.

**Strengths:**

- The proposal of a legal benchmark, using real data, for evaluation law-experts agents seems quite relevant.

- It is nice that a LLM-based MAS system is proposed and evaluated on that benchmark.

- The paper is in general well-written and well-presented.

**Weaknesses:**

- One of the key contributions seem to be the proposal of a benchmark dataset. However, currently the evaluation of that dataset itself seems to be quite poor. It is only evaluated by 3 students, and they see a small percentage of the dataset (30 questions out of 950). The data may even have some anomalous behaviour (such as one student simply giving 100 to every criteria).

If the dataset itself may have issues, then it is hard to reach conclusions based on that benchmark.

- I wonder a bit about the option of using multiple choice questions. Surely a team of legal agents would not be used to answer multiple choice questions, but rather to take a legal decision/recommendation. Hence, perhaps there is a gap between the dataset and the actual intended application?

- Some of the conclusions of the paper do not really seem to follow from the experimental results. Additionally, there is no analysis of the variance or the statistical significance, which makes it even harder to accept the conclusions (e.g., "44 out of 60 top results are achieved under our designed MAS", while there are several values in the table quite close to each other).

Detailed comments:

- "Unfortunately, few studies have explored the potential of MAS in legal tasks," -> Which studies have done that before? Citation missing?

- "IRAC reasoning is designed to address the limitation of classical deductive reasoning, where the truth of the premises in a legal argument is often neither straightforward nor self-evident1 ." -> Why cite a book as a footnote?

- "Intuitively, we define the following mapping relations to bridge the actual data with our conceptual framework discussed in Section 3:" -> Somewhat strange that the table is not presented in a table environment, but just follows directly in the text.

- "We invited three students with legal backgrounds or prior experience in legal-related research." -> Is that good enough? May need to scale up the human evaluation? E.g., evaluator 2 simply gives 100 to everything?

- "A total of 30 samples were randomly selected, each including the original text, the extracted question, and the corresponding answer." -> 30 out of 950? It doesn't seem to be good enough?

- Table 2: I think the performance of Random @3 and @5 could also be calculated?

- Table 2, Qwen2.5-7B Instruct: Wrong formatting, CS is the 2nd best (62.84), not F+LR+AR+CS (62.74).

- "Richer contexts can lead to better performance. The results indicate that involving more agents and providing richer reasoning steps generally leads to improved performance. For instance, ’BM25@5’ outperforms ’BM25@3’ when using the GPT-4o-mini model." -> I don't see how you reach this conclusion by simply changing from @3 to @5, you are just increasing the output ranking size, which should obviously increase the results monotonically in any case.

Additionally, there are several cases where a lower number of agents leads to better results. For instance, in the @1 column, there is only one case where the highest number of agents has the highest result.

- "This effect is more pronounced in larger-parameter models, such as DeepSeek-v3.1 and GPT-4o-mini, suggesting that the improvements brought by MAS enable the Meta-LLM to better evaluate the execution results of these agents." -> It is hard to see this conclusion from your table. For instance, several cases in DeepSeek only 2 agents lead to the top results.

- "Our results show that 44 out of the 60 top performances (bold values) are achieved under our designed MAS, demonstrating the effectiveness of our MAS design as well as its potential for legal tasks." -> What about statistical ties? In how many cases are you really statistically significantly better?

- "The best performance is often achieved when agents handling Legal Rules or Common Sense are activated." -> Curiously, many times the results drop when CS is activated (from second to last row to last row of every block).

- "We conduct a case study on this phenomenon to investigate the underlying reasons, as illustrated in Figure 3." -> Table 3.

- Section 7 is quite strange, shouldn't that be in related work?

- Ethics: human annotators were used, but it was not mentioned whether this work obtained ethical approval.

- "To ensure the reproducibility of our experimental results, we put our detailed implementations under Section 5." -> Not sure I follow, did you mean to say that your methodology is explained in detail?

- Appendix: "with a average of approximately 185.53 chunks." -> "an average"

**Questions:**

- In how many cases the MAS system is actually statistically significantly better than the single agent system?

- Why an evaluator has 100 for every criteria? Can the data be trusted?

- Did you obtain ethical approval for this work?

- As shown in my detailed comments, in several places I struggle to reach your conclusions based on your experimental data. Can you clarify?

**Details Of Ethics Concerns:**

Human annotators are used, but the paper does not mention whether ethical approval was obtained by their institution.

---

### Official Review · Reviewer_MqBR · 2025-10-30

**Soundness:** 2
**Presentation:** 2
**Contribution:** 1
**Rating:** 2
**Confidence:** 4

**Summary:**

This paper proposes legal benchmark collected from GDPR related documents consisting of 950 questions and various context for answering these questions. It also proposes an multi-agent framework, which first decomposes a question into atomic subtasks and then solve them step by step. To answer the subquestions, they assign agents different roles, based on the problem solving procedure in legal domain.

**Strengths:**

The writing is clear and easy to follow
Since there aren’t many LLM benchmarks in the legal domain, this dataset could be a useful addition.

**Weaknesses:**

The main contribution of the paper seems to be the dataset. But there are several issues:
1. The dataset is rather narrow. It only focuses on GDPR-related questions. This feels too limited to really assess how well LLMs handle legal tasks in general.
2. No comparison with other legal benchmarks. The paper doesn’t show how this dataset stands out from existing ones. The authors claimed the result dataset is the only one with rich context. But there are similar datasets for comparison, for example, "LEXam: Benchmarking Legal Reasoning on 340 Law Exams"
3. Dataset quality isn’t clearly validated. The dataset is created in a semi-automatic way (using LLMs to extract questions), but only 30 samples are checked manually. There’s also no information on how annotations were done, what criteria were used, or whether annotators agreed with each other.

The proposed multi-agent framework MAS lacks novelty. The core idea is to decompose questions into subquestions, which has been extensively explored in previous works. Besides, the experiments don’t reveal much. The only real takeaway is that more context leads to better performance, which is pretty expected. Therefore, it’s still not clear how this dataset helps us better understand LLMs’ legal reasoning abilities.

**Questions:**

Please see the weaknesses.

---

### Official Review · Reviewer_RDkk · 2025-11-01

**Soundness:** 2
**Presentation:** 2
**Contribution:** 2
**Rating:** 2
**Confidence:** 4

**Summary:**

The authors present a novel benchmark for the legal domain, focusing on multi-agent systems (MAS). The authors fill a gap in benchmarks for MAS in the legal domain. Multi-agent systems show great promise, and the legal domain would inherently profit greatly from multi-agent solutions due to natural task decomposition, e.g., the popular IRAC method. In the reviewer's opinion, the primary contribution of the paper is the dataset comprising 950 legal questions extracted from official GDPR-based court cases. The authors also propose a framework for multi-agent systems for the legal domain, based on a popular legal task decomposition called IRAC. Lastly, they present results for different frontier models.

**Strengths:**

# Originality
- A benchmark in the legal domain, with multi-agent systems in mind, seems useful to the community. The task composition into IRAC appears practical and well-motivated.

# Quality
- The data collection and filtering process appears reasonable, and the authors attempt to validate the dataset through human annotations.
- The authors demonstrate their results transparently and include a range of analyses of both the dataset and frontier model performances.

# Clarity
- The paper is overall easy to read.
- Tables and Figures are well formatted.

# Significance
- A well-constructed legal benchmark with a focus on task decomposition suitable for multi-agent systems could provide a valuable signal for method development.

**Weaknesses:**

## Quality
1. Three annotators for 30 samples (unclear if each got the same 30 samples, 10 samples per annotator?) is not a particularly extensive human annotation effort. Having worked with human annotators extensively, the reviewer's experience is that at least 2 annotations per sample, ideally 3 as a tie breaker, are required to get a notion of inter-human agreement to validate the quality of any human annotations, even in specialist domains (This is from personal experience; the reviewer couldn't find a good citation for this claim). It would then be useful to analyse where the human annotators disagree. While a 30-sample annotation round can provide an early signal, it is the reviewer's opinion that it does not suffice to paint the whole picture on the annotation quality and thus the verification of the dataset creation. (1) Please clarify the annotator setup in more detail, and potentially provide inter-human agreement, and provide some examples where human annotators disagreed with either the model's extraction or with other human annotators. (2) Would it be feasible to get more human annotations? (3) Can you provide more details about the randomly selected samples? Case 14 is dominating the dataset. Are most samples coming from Case 14?
2. The presented results, e.g., Table 2 and 3, Figure 2 and 3, don't contain any measure of variance. The experiment description does not mention any hyperparameters or whether these results were run over multiple seeds. It is the reviewer's opinion that all claims of Sections 6.1 and 6.2 are not supported sufficiently. (4) Please add some measures of variance, e.g., standard deviation or standard error, for at least some models. Please report your hyperparameters for all models, such as temperature, context length, etc.
3. The results for EMB are reported but never discussed. They also seem to suggest that LR and F+LR often perform better than any MAS, as evidenced by the GPT-4o-mini results. (5) When the authors get results with more statistical significance, can they also discuss the EMB results?


## Clarity
1. The abstract is wordy and could be improved by leaving out some adjectives. For example, change

> Our benchmark uses GDPR as the application scenario, encompassing extensive background knowledge and covering complex reasoning processes that effectively reflect the intricacies of real-world legal situations.

into

> Our benchmark uses GDPR as the application scenario, encompassing background knowledge and covering
reasoning processes that reflect the intricacies of real-world legal situations.

2. The Introduction is slightly hard to read, with the introduction of "reasoning steps" as "sub-task" (line 54), which then become "subproblems", and the subproblems are passed to the MAS, even though in the pseudocode, the full case description is passed to the Meta-LLM, which is part of the whole MAS. Either way, clarification of that paragraph would greatly improve the flow.
3. Could you please clarify if you were referring specifically to the legal domain with the following sentence (line 62):

> To the best of our knowledge, this is the first benchmark that provides sufficiently rich context to enable multiple LLM agents to collaborate in reasoning and exploration

4. The unfolded enumeration is hard to read and easily improvable in Section 2.3
5. Line 158 typo: "when an MAS" -> "when a MAS"
6. Could the authors clarify the use of bookmarks? Why don't the authors use citations here, especially when citing a book? Additionally, when citing a book, please specify the exact pages you're referring to.
7. It is the reviewer's opinion that Figure 1 would probably look better on page 3. Currently, it's two pages removed from its first reference.
8. In Section 3.2, the header of "1)" does not end with a period, but the headers of "2)" and "3)" end with a period.
9. In Section 3.2, Algorithm 1 is inconveniently placed after Step 2, not right after it was referenced in Step 1.
10. What are "reality issues" in Section 4.2?

**Questions:**

1. Could you please clarify what the difference between a legal decision and a legal opinion is, and who decided which sample belongs to which category? This distinction was not validated in the human annotations either. Who annotated the samples into legal decisions and legal opinions?
2. Could you please cite BM25 when it's first referenced?
3. Table 4 does not really convey a lot of information. For example, taxonomies are not discussed anywhere else in the text, so it's unclear what exactly is being referred to. Data type must have a type for "Hybrid". It is also unclear how "Task Decomposition" would be formally defined in this context.

To summarise, the reviewer believes that all the necessary components are in place for a potentially strong contribution and a robust benchmark. However, each component still lacks quality and depth. First, the performance analysis focuses only on accuracy, without any measure of variance. Are some tasks harder than others? Are there other statistical measures for a dataset with both yes/no and single-choice questions that might be interesting? Currently, F+LR are hardcoded. Could the MAS extract the relevant sections itself, given the source document? Second, the human annotation effort is a starting point, but it does not reach a level that would provide confidence in the quality of the entire dataset. Third, the paper's formatting appears rushed and requires improvement.

The reviewer is looking forward to the discussion during the rebuttal period.

---

### Note · Authors · 2025-12-03

**Comment:**

We advise that the current version of the manuscript requires substantial revision to fully address the concerns raised by the reviewers. We plan to incorporate the valuable feedback we received and submit an improved version to a future venue.

We would like to thank the reviewers and the AC for their constructive comments and their time.

**Withdrawal Confirmation:**

I have read and agree with the venue's withdrawal policy on behalf of myself and my co-authors.